# Impact of Polypharmacy for Chronic Ailments in Colon Cancer Patients: A Review Focused on Drug Repurposing

**DOI:** 10.3390/cancers12102724

**Published:** 2020-09-23

**Authors:** Riccardo Giampieri, Luca Cantini, Enrica Giglio, Alessandro Bittoni, Andrea Lanese, Sonia Crocetti, Federica Pecci, Cecilia Copparoni, Tania Meletani, Edoardo Lenci, Alessio Lupi, Maria Giuditta Baleani, Rossana Berardi

**Affiliations:** Division of Medical Oncology, University Hospital—Marche Polytechnic University, Ancona, 60126 Marche, Italy; lucacantini.med@gmail.com (L.C.); chica.giglio@gmail.com (E.G.); alessandro.bittoni@ospedaliriuniti.marche.it (A.B.); andrea.lanese@ospedaliriuniti.marche.it (A.L.); crocetti.sonia@alice.it (S.C.); peccifede91@gmail.com (F.P.); cecicoppa@hotmail.it (C.C.); tania.meletani@gmail.com (T.M.); edoardo.lenci@hotmail.it (E.L.); lupialessio2@gmail.com (A.L.); mg.baleani@hotmail.it (M.G.B.); r.berardi@staff.univpm.it (R.B.)

**Keywords:** colon cancer, drug repurposing, NSAIDs, antihypertensive, metformin, antidepressants, statins, antibiotics

## Abstract

**Simple Summary:**

Colorectal cancer patients are frequently also affected by various chronic conditions that require specific treatment. Several papers have focused on the role of different classes of drugs such as anti-hypertensive medications, statins, anti-bacterial antibiotics, aspirin and NSAIDs, metformin and anti-depressants, and their impact on colorectal cancer survival. Aim of this review is to summarise this findings as to suggest which drugs might be further explored for therapeutic approaches in this setting. Our review suggests that beta-blockers and statins should be further explored as potentially useful treatment options respectively in metastatic colorectal cancer for the former and in adjuvant setting for the latter.

**Abstract:**

Colorectal cancer is characterized by high incidence worldwide. Despite increased awareness and early diagnosis thanks to screening programmes, mortality remains high, particularly for patients with metastatic involvement. Immune checkpoint inhibitors or poly (ADP-ribose) polymerase (PARP)-inhibitors have met with disappointing results when used in this setting, opposed to other malignancies. New drugs with different mechanisms of action are needed in this disease. Drug repurposing might offer new therapeutic options, as patients with metastatic colorectal cancer often share risk factors for other chronic diseases and thus frequently are on incidental therapy with these drugs. The aim of this review is to summarise the published results of the activity of drugs used to treat chronic medications in patients affected by colorectal cancer. We focused on antihypertensive drugs, Non-Steroid Anti-inflammatory Drugs (NSAIDs), metformin, antidepressants, statins and antibacterial antibiotics. Our review shows that there are promising results with beta blockers, statins and metformin, whereas data concerning antidepressants and antibacterial antibiotics seem to show a potentially harmful effect. It is hoped that further prospective trials that take into account the role of these drugs as anticancer medications are conducted.

## 1. Introduction

Despite advances in treatment and diagnosis, cancer represents one of the major causes of death in the world. The prognosis for patients suffering from other chronic conditions such as obesity, diabetes and hypertension is usually far more favourable than incurable cancer.

Due to frequent “sharing” of the same risk factors, many patients suffering from chronic conditions that also have high risk of developing a series of malignancies. Smoking habit can be considered the most relevant risk factor for the majority of chronic diseases as well as the majority of cancer types.

Most clinical trials usually enrol patients with less relevant comorbidities and when the impact of these is negligible on patient’s performance status: because of that, we have almost no data concerning the different roles of oncological treatments in patients who are already under other medications for other pre-existing conditions or whether these drugs can have positive or harmful effects on cancer specific survival.

Colon cancer is one of the cancer types with the highest prevalence, and new drug development has focused either on the anti-Epidermal Growth Factor Receptor (EGFR) or anti-Vascular Endothelial Growth Factor (VEGF) pathways [1,2]. New therapeutic approaches such as immunotherapy with checkpoint inhibitors seem to have met with disappointing results in this kind of disease, particularly if compared with other high-incidence cancer types such as lung cancer. Colon cancer patients also frequently have a series of comorbidities that require chronic treatment. This is usually caused by the fact that colon cancer shares many risk factors with other chronic conditions.

Recently, there has been renewed interest in the concept of drug repurposing: many drugs that were first used in a setting have been proven to be equally effective, or perhaps even more effective, in treating other conditions.

One notable example is the one of amantadine: the drug was first used as medication to treat type-A influenza [3], but it has been demonstrated that it is also effective for the treatment of diskinesia due to Parkinson’s disease, and actually, this has become its main indication [4].

Thalidomide is another example: it was first used to treat morning sickness and “anxiety” during pregnancy, but after the discovery that it led to birth defects, its use was stopped [5]. It has been recently “rediscovered” to treat a series of different conditions such as leprosy and multiple myeloma in oncology [6].

Drug repurposing is a particular field of research in medical oncology: costs usually associated with the discovery of new molecules and phase I trial testing could be completely avoided should we find that old drugs that have already been approved for other indications can be beneficial also to treat different types of malignancies.

The aim of this review is to summarise what kind of data has been actually published and presented on this topic concerning patients affected by colon cancer. The different sections of this review are based on the type of drugs on which the section will focus.

## 2. Results

### 2.1. Antihypertensive Drugs

Primary hypertension is one of the most common chronic conditions worldwide, particularly in western countries. There is actually no cure for hypertension; however, a wide number of drugs have been used to treat this disease.

Most active drugs focus on peripheral vascular resistances, one of the key factors causing increases in blood pressure. Drugs such as angiotensin-converting enzyme inhibitors (ACEi); angiotensin II receptor blockers (ARB); calcium channel blockers (CCBs); and alpha-blockers, also known as alpha-adrenoreceptor antagonists, are active in decreasing blood pressure by reducing peripheral resistances through reduced blood vessels’ smooth muscle contraction.

Beta blockers, also known as beta-adrenoreceptor antagonists, oppositely decrease blood pressure and heart pulse rate via a negative inotropic effect on the heart, thus decreasing left ventricular ejection fraction.

These drugs have been extensively used in patients with colon cancer, particularly when patients receive anti-VEGF-based chemotherapy combinations that might lead to increased blood pressure. Development of high blood pressure in metastatic Colorectal Cancer (mCRC) patients treated with anti-VEGF drugs has been described as a surrogate of better outcome for treated patients [7,8], thus leading to difficult treatment choices when deciding to lower blood pressure.

A series of papers have assessed whether prolonged use of antihypertensive drugs can lead to increased incidence of different cancer types:

Bangalore et al. [9] reported the results of a network meta-analyses enrolling 324,168 patients treated with different antihypertensive drugs and assessed the risk of cancer development. Increased cancer risk was not observed either for users of ARB (odds ratio (OR): 1.01, 95% confidence interval (95% CI): 0.93–1.09), ACEi (OR: 1.00, 95% CI: 0.92–1.09), beta blockers (OR: 0.97, 95% CI: 0.88–1.07), CCBs (OR: 1.05, 95% CI: 0.96–1.13) and diuretics (OR: 1.00, 95% CI: 0.90–1.11). Only 2% patients ultimately developed cancer. When ACEi was used together with ARB, there was a greater than 10% increase in the relative cancer risk yet no difference in survival could be demonstrated. This meta-analysis did not take into account different types of cancer, but rather, it was focused only on assessing the overall risk of cancer development and mortality.

The ARB trialists group [10] conducted a meta-analysis based on 15 trials, enrolling 138,769 patients treated with different ARBs: candesartan, losartan, valsartan, irbesartan and telmisartan. Incidental cancer rate was 6.16% of ARB users vs. 6.31% of non-ARB users. There were no statistically significant differences when individual ARBs were assessed (OR: 1.00, 95% CI: 0.95–1.04). The most frequent cancer types were lung, prostate and breast cancer, thus leaving the number of patients that were diagnosed with colon cancer relatively small.

Recently, Htoo et al. [11] reported the results of a new-user cohort study of U.S. Medicare beneficiaries aged over 65 who received antihypertensive treatment with ARB/ACEi vs. other guideline-approved medications. In 111,533 patients, 532 newly diagnosed colorectal cancer were found, about 5 per thousand. There was no increased risk of CRC cancer development when comparing ARB/ACEi vs. beta blockers (hazard ratio (HR): 1.0, 95% CI: 0.85–1.11), ARB/ACEi vs. CCBs (HR: 1.2, 95% CI: 0.97–1.4) and ARB/ACEi vs. thiazides (HR: 1.0, 95% CI: 0.80–1.3). This was proven either for 2 or 5 years of treatment.

Sorensen et al. [12] reported the impact of CCB use in 23,167 patients in terms of increased cancer risk and mortality: 967 cases of cancer were diagnosed (4%), and there was no increased risk of colon and breast cancer. Cancer mortality was close to what is expected in the large population (standardized mortality ratio: 0.97, 95% CI: 0.89–1.04).

Assimes et al. [13] conducted a nested case control study of patients who developed cancer and who received either beta blockers, CCBs, ARB/ACEi or thiazides; 11,697 cancer patients were matched up to 10 controls each. Most patients received antihypertensive drugs for a timespan ranging from 3.6 to 5.7 years. The most common cancer types were colon, head and neck, lung and haematological cancer. There was no difference in either incidence or mortality for long-term or short-term users (upper bound of 95% CI of HR: 1.45 for short term users).

These results suggest that neither long- or short-term exposure to antihypertensive medications can lead to an increase or decrease in colon cancer incidence or mortality.

On the other hand, there are a series of papers that reported markedly different outcomes for patients who have already developed colon cancer and who are on treatment with different antihypertensive drugs. Most of papers have focused on large registry analyses, with a few studies focused on specific clinical presentations:

Cui et al. [14] reported survival outcomes of 2891 patients with incident breast, colorectal, gastric and lung cancer and associations with common antihypertensive drugs. ARB users with colorectal cancer seemed to have better overall survival (OS) (HR: 0.62, 95% CI: 0.44–0.86) and disease-specific survival (DSS) (HR: 0.63, 95% CI: 0.41–0.98). Beta-blocker users with colorectal cancer also seemed to have better OS (HR: 0.50, 95% CI: 0.35–0.72) and DSS (HR: 0.50, 95% CI: 0.34–0.73).

In a preliminary study of Jansen et al. [15], it was reported that, among 1975 patients diagnosed with CRC from 2003 to 2007, 28% were beta-blocker users. OS was not significantly different comparing beta blocker users vs. those who were not on concomitant treatment with beta blockers (HR: 0.99, 95% CI: 0.79–1.22), and cancer specific survival (CSS) was also not significantly different (HR: 0.93, 95% CI: 0.71–1.21). When stratifying for stage at diagnosis, however, a statistically significant better OS was seen for patients with stage IV involvement and diagnosis (HR: 0.50, 95% CI: 0.33–0.78, *p* = 0.0033) and CSS was better (HR: 0.47, 95% CI: 0.30–0.75, *p* = 0.0017); 256 stage IV patients were included in the analysis, and 27% were beta-blocker users. More than 95% of the patient population received surgery, about 40% patients received chemotherapy, while only 2–4% of patients received either anti-VEGF or anti-EGFR drugs. Median survival improvement was around 17 months.

In a following study, Jansen et al. [16] performed the same analysis on 8100 patients; 22% were beta-blocker users before diagnosis of colorectal cancer. Pre- and post-diagnostic beta-blocker usage seemed to not have an impact on survival (HR: 1.07, 95% CI: 0.96–1.19 and HR: 1.10, 95% CI: 0.98–1.23 respectively for pre- and post-diagnostic use). Cumulative use of beta blockers (1–12 months) was associated with an increase in mortality (HR: 1.20, 95% CI: 1.03–1.39).

Giampieri et al. [17] conducted a retrospective analysis on 235 patients with metastatic colorectal cancer, treated with first-line chemotherapy doublet or doublet plus bevacizumab, stratified by concomitant beta blocker use; 12% of the patient population was on treatment with beta blockers. In patients treated with chemotherapy, worse overall survival was observed for patients who did not receive beta blocker vs. those who did (median overall survival (mOS): 25.7 vs. 41.3 months, HR: 2.25, 95% CI: 1.04–3.42, *p* = 0.03). In patients treated with chemotherapy and bevacizumab instead, those who did not receive beta blockers had better survival vs. those who were on incidental beta-blocker treatment (mOS: 23.6 vs. 18.5, HR: 0.89, 95% CI: 0.38–2.07, *p* = 0.77).

Fiala et al. [18] reported the clinical outcome of 514 mCRC patients treated with bevacizumab, stratified by incidental use of beta blockers; 126 patients (24%) were incidental beta-blocker users. Median progression free survival (mPFS) for beta-blocker users vs. not were respectively 11.4 vs. 8.3 months (*p* = 0.006), whereas mOS were respectively 26.8 vs. 21.0 (*p* = 0.009). Multivariate analysis confirmed an independent prognostic role for beta-blocker use both for progression free survival (PFS) (HR: 0.763) and OS (HR: 0.73). Interestingly, other antihypertensive drugs seemed not to have a prognostic impact on both PFS and OS.

Cardwell et al. [19] also reported survival outcomes for a series of patients who developed breast, colorectal and prostate cancer, stratified by different types of antihypertensive medication that were prescribed. Focusing on 1511 patients with colorectal cancer treated with ARB/ACEi vs. 7291 colorectal cancer controls treated with other antihypertensive drugs, there was no increase for cancer-specific mortality (OR: 0.82, 95% CI: 0.64–1.07 and OR: 0.78, 95% CI: 0.66–0.92 respectively for ARB and ACEi).

The studies that focused on antihypertensive drugs and their impact on prognosis for patients who already developed colorectal cancer are listed in the following Table 1.

Summing up these results, it is suggested that beta-blocker users might have better survival, albeit disease stratification for other recently proven prognostic features such as tumour sidedness or presence of B-raf/K-ras/N-ras mutations are lacking in all these studies. Other drugs used to treat hypertension seem not to have the same positive prognostic impact.

There is no definitive explanation of the mechanisms that underlie this findings: even though a series of preclinical data have been published on the matter [20,21,22], these studies have stopped in the preclinical phase, failing to have some form of validation in the clinical setting.

Another explanation of these findings can be traced back to surgery as a confounding factor: it has previously been described that beta blockers might reduce morbidity and mortality after surgery, and there are some published data that suggest that this might also prove true for colorectal cancer patients submitted to emergency surgery [23].

To conclude, drugs used to counter hypertension in patients with colorectal cancer seem not to have a negative prognostic impact on patient survival. Beta blockers, among all other forms of antihypertensive medications, should be further investigated as candidates for new treatment options for these patients.

### 2.2. Non-Steroid Anti-Inflammatory Drugs (NSAIDs)

Arguably, in addition to high blood pressure, atherosclerosis accounts for most chronic conditions affecting the cardiovascular system.

Patients suffering from heart ischaemia or other conditions that can be associated with atherosclerosis are usually treated with non-steroid anti-inflammatory drugs (NSAIDs), as their effect reduces platelet aggregation and their positive effect prevents further damage upon relapse of ischaemic episodes.

Correlation between NSAID use and the risk of developing colon cancer has been largely investigated in the literature. There are indeed many papers focused on aspirin (acetylsalicylic acid, ASA) and cancer:

Daily assumption of ASA seems to have a protective role on colon cancer tumourigenesis by several mechanisms of action. Early inhibition of inflammation and promotion of specific antitumour immunity may play a preventative role in colorectal cancer development [24,25,26].

Another proposed mechanism that might explain favourable antitumour effects is the one related to the antithrombotic properties of ASA via reduced interaction between platelets and cancer cells. Platelets have still an unknown role in cancer progression; however, different studies suggest that they could be implicated in enhancing tumour proliferation and metastatic spread [27,28,29,30].

The chemopreventive action of ASA could be explained through the antiproliferative effect mediated by inhibition of the cyclooxygenase (COX) enzyme COX-2 and the transcription factor c-MYC [31,32,33]. Different studies have demonstrated that the oncoprotein c-MYC is frequently activated in tumour cells (either in colon cancer or in other tumour types). Activation depends on microenvironmental signals mediated by the release of growth factors initially stored inside platelets when these come into contact with circulating tumour cells (CTCs) [34,35].

Other studies suggest that activated platelets could mediate immune evasion contributing to cancer metastatic diffusion [36,37].

On these bases, Mitrugno et al. [38] showed that the daily use of low-dose aspirin could reduce the proliferation of tumour cells through inhibition of platelet-derived signals required for the upregulation of the oncoprotein c-MYC.

Actual clinical data related to the role of ASA use in chemoprevention of colon cancer comes from a series of clinical trials: the majority of them suggest that long-term treatment with low doses of aspirin can decrease the incidence of de novo adenomas and colorectal cancers [39]. Based on these data, use of ASA for chemoprevention of colorectal cancer in subjects aged 50 to 59 is recommended in the United States, albeit this cannot be considered a suitable alternative to colorectal cancer screening.

Risk–benefit assessment concerning use of non-aspirin-NSAIDs (NA-NSAIDs) for colorectal cancer prevention is still unknown: clinically relevant protective effects of NA-NSAIDs use have been found in women (19% relative risk reduction), distal colon cancer (22% relative risk reduction) and white people (risk reduction from 31% to 41%) [35].

Indomethacin (NA-NSAIDs) seems to be particularly effective in reducing risk of colorectal cancer occurrence.

Recently, the potential antiproliferative effect in colorectal cancer of juglone in combination with indomethacin was investigated: the study showed superiority of the two-drug combination, whilst treatment with juglone alone was not as effective as with indomethacin [40].

Jiang et al. [41] assessed the benefit of adding ASA to cisplatin in colorectal cancer cell lines. The study showed that, in vitro, ASA significantly enhances the cisplatin-mediated inhibition of cell proliferation, migration and invasion and induces apoptosis in colon cancer cells. The combined treatment of aspirin and cisplatin suppresses the expression of the antiapoptotic protein Bcl-2 and the EMT-related proteins, upregulates the levels of the cleaved PARP and Bax, blocks the PI3K/AKT and RAF-MEK-ERK signalling pathways, and inhibits the binding activity of NF-κB to the COX-2 promoter. Therefore, the combination of aspirin and cisplatin downregulates COX-2 expression and PGE2 synthesis.

When it comes to patients who have already developed colorectal cancer, data concerning clinical usefulness of NSAIDs are rarer; use of aspirin seems to improve survival particularly in patients with colorectal cancer harbouring PIK3CA mutations [27,28,29,30,31,32,33,34,35,36,37,38,39,40,41,42].

Giampieri et al. [43] reported the outcome of patients with metastatic colorectal cancer who received capecitabine rechallenge after failure of other palliative treatment options, stratifying patients by ASA concomitant use. The analysis was conducted prior to introduction in everyday clinical practice of regorafenib and TAS-102. The study suggested that ASA concomitant use might be associated with better PFS (6.5 months for patients receiving treatment with aspirin versus 3.3 months for patients who were not on aspirin, *p* = 0.0042) suggesting that ASA may improve the clinical outcome in this setting of patients.

Restivo et al. [44] reported outcomes for patients with locally advanced rectal cancer who received neoadjuvant chemoradiotherapy, stratified by concomitant ASA use. ASA users had greater overall radiological response rate (67.6% vs. 43.6%, *p* = 0.01), greater pathological response (46% vs. 19%, *p* < 0.001), better 5-years disease free survival rate (86.6% vs. 67.1%, HR: 0.20, 95% CI: 0.07–0.60) and 5-years overall survival rate (90.6% vs. 73.2%, HR: 0.21, 95% CI: 0.05–0.89) compared to nonusers. Improvement in disease-free survival and overall survival was mainly due to reduction in lower rate of distant metastases development (HR: 0.30, 95% CI: 0.10–0.86).

The studies that focused on aspirin and NSAIDs and their impact on colorectal cancer survival or antiproliferative effect are listed in the following Table 2.

In conclusion, literature data suggests that NSAIDs may have a key role in colon cancer prevention; they could also influence response to chemotherapy and therefore patients’ prognosis. There are currently ongoing clinical trials (mostly focused on colon cancer patients harbouring PIK3CA mutations) that will look into this matter in detail.

### 2.3. Antibacterial Antibiotics

Antibacterial antibiotics are one of the classes of drugs most commonly used. Usually these drugs are overprescribed compared to real clinical necessity. On the other hand, there are some chronic conditions, such as chronic obstructive pulmonary disease, that frequently require antibiotic treatment due to frequent bacterial superinfection.

It is suggested by some authors that antibiotics use could increase risk of cancer development:

Petrelli et al. [45] published a recent metanalysis comprising 7,947,270 participants from 25 observational studies. The study showed that antibiotic use was associated with an 18% relative risk increase of cancer. The highest risk was found in patients with longer duration of treatment or higher doses of antibiotics exposure. The study showed 30% increased incidence of lung, haematological, pancreatic and genitourinary cancer compared to controls and a small increase in risk for CRC or gastric cancer. The authors’ hypothesis is that antibiotics may change gut microbiota and finally its interaction with immune system, ultimately leading to reduced immunosurveillance.

Colon-rectal cancer is well documented to be related with alteration of the gut microbiota in preclinical studies [46,47].

Kaur et al. [48] showed that long-term antibiotic use may cause depletion of the natural bacterial flora, leading to a reduced number of mucous-producing goblet cells. This can lead to chronic bowel inflammation, which is a crucial factor in tumour development and progression.

Another preclinical study [49] highlighted the active role of microbiota on the progression of intestinal adenomas to tumour: APCmin/* mice gavaged by faeces of CRC patients showed progression of intestinal adenomas compared to those fed with faeces from healthy controls.

Similar differences in terms of active gut microbiota species were seen in humans affected by CRC compared with healthy volunteers [50].

Some authors speculate that the alteration of the gut microbiota caused by antibiotics exposure leads to depletion of anti-inflammatory and short-chain fatty acid, producing an increase in pro-inflammatory bacteria and altering immune response [51,52].

A nested case-control [53] reported that, comparing anti-anaerobic vs. anti-aerobic antibiotics, the risk of CRC development was greater with the former. This was explained as being related to the fact that the gut microbiota is predominantly composed of anaerobes. The risk was dose-dependent and independent from other prognostic factors such as sex, age, comorbidities and co-medication. However, when applying a stratification factor by different classes of antibiotics, only penicillin and quinolone uses were significantly associated with increased risk.

Zhang et al. [54] confirmed these findings, particularly in the colon proximal tract, in another case control study. When they assessed the risk of rectal cancer development, an inverse association of antibiotic use vs. nonusers was found (*p* = 0.003). Penicillin, particularly the ampicillin/amoxicillin combination, increased the risk of colon cancer (adjusted OR: 1.09, 95% CI: 1.05–1.13), whereas tetracycline use reduced the risk of rectal cancer (adjusted OR: 0.90, 95% CI: 0.84–0.97).

The microbiota alteration caused by antibiotics has also been demonstrated to be associated to changes in chemotherapy effectiveness:

A preclinical study [55] showed that the microbiota dysbiosis caused by antibiotics administration in mice affected by CRC significantly reduced the cytotoxic effect of 5-FU on tumour growth compared with mice treated only with 5-FU. It was hypothesised that this could be traced back to increases in the number of pathogenic bacteria, while beneficial bacteria contributing to anti-tumour activity were decreased. In particular, the species that would help Treg differentiation through production of short-chain fatty acids, that would be decreased by wide-spectrum antibiotic therapy, were considered responsible for this reduced efficacy.

These findings were confirmed in humans: Abdel-Rahaman et al. [56] demonstrated that antibiotic use before, but not following, the start of 5FU-based chemotherapy was associated with worse progression-free survival (*p* = 0.001) and worse overall survival (*p* < 0.001) in patients with metastatic colorectal cancer. However, these findings need to be validated within a larger dataset of patients, as the authors suggest.

Despite the bulk of papers that have been published on the matter seem to suggest negative prognostic role of prior antibiotic use in patients with metastatic colorectal cancer, there are also some published data that seem to come to different conclusions:

Hekmatshoara et al. [57] reported that administration of antibiotics reduced the number of β glucuronidases producing bacteria which can convert the inactive metabolite SN-38G to the active and toxic SN-38, suppressing irinotecan cytotoxicity such as diarrhoea. It was also reported that the *Fusobacterium nucleatum* can decrease the CRC cell chemosensitivity to 5-FU both in vitro and in vivo, finally causing chemoresistance in patients with advanced CRC. Moreover, these bacteria promote a pro-inflammatory microenvironment in the colon, suppressing host immunity with consequent increase of tumour growth and playing a role even in CRC recurrence. Therefore, both from a theoretical and preclinical point of view, eradication of *Fusobacterium* spp. through the use of active antibiotics should have a positive impact on patient’s response to treatment and survival.

To conclude, despite the majority of studies seemingly suggesting a negative prognostic impact of antibiotic use on patients’ prognosis, particularly for antibiotics with a wide spectrum of activity such as penicillin or quinolone, the influence of antibiotics on CRC development as well as on chemotherapy efficacy is still under investigation.

### 2.4. Antidepressants

Antidepressant agents are commonly used in cancer patients for treatment of anxiety, depression, insomnia and other mood disorders. The principal classes are Tricyclic (TCAs), Monoamine Oxidase Inhibitors (MAOIs), Selective Serotonin Reuptake Inhibitors (SSRIs), Serotonin-Noradrenaline Reuptake Inhibitors (SNRIs) and other agents such as mirtazapine. A lot of trials have evaluated the relationship between the use of antidepressants and colorectal cancer in preclinical and clinical settings.

In 2006, Arimochi et al. [58,59] have shown that, in human HT29 and HCT116 colon carcinoma cells, TCAs reduce cell viability in a concentration-dependent manner and cause apoptotic cell death through either a non-mitochondrial or a mitochondrial pathway. Recently, Begona et al. [60] demonstrated the anti-invasive and antimetastatic properties of imipramine thanks to the anti-Fascin 1 activity in serrated colon adenocarcinoma cells.

Several studies have demonstrated the anticancer activity of SSRIs in colon cancer cells and animal models.

In particular, sertraline and paroxetine may induce a dose-dependent inhibition of cell viability and proliferation comparable to cytotoxic agents such as 5-fluorouracil. They also arrest cells at the G0-G1 stage and stimulate DNA fragmentation in a dose-dependent manner [61].

Moreover, paroxetine induces cell apoptosis through MET and ERBB3 inhibition, leading to the induction of the JNK and caspase-3 pathways and the reduction of the expression of antiapoptotic protein Bcl-2 [62,63]. On the other hand, fluoxetine showed an antiproliferative effect on colon cancer cells by the reduction of VEGF expression [64] and by the alteration of tumour-related energy generation machinery [65], especially under conditions of hypoxia. Recently, Marcinkute et al. confirmed fluoxetine’s cytotoxicity in human colon cancer cells through p53-independent apoptosis [66]. In addition, in 2015, Iskar et al. demonstrated that citalopram reduces the number of circulating tumour cells and the number of metastases in distant organs in mice, through inhibition of TGF-β signalling [67].

In 2012, Fang et al. showed that, in animal models, mirtazapine inhibits tumour growth through immune system activation, testified by higher serum levels of IL-12, CD4+ and CD8+ and lower tumoural concentrations of TNF-α and IFN-γ [68].

When these promising suggestions moved on to clinical testing, a few contradictory results were found:

In 2006, Xu et al. [69] reported 30% reduced risk of CRC among users of high doses of SSRIs, and no consistent relationship was recorded for the risk of CRC and the use of TCAs. Before this study, there are no other epidemiologic data on the relation of antidepressant use to the risk of CRC.

Coogan et al. [70] also reported an inverse association of risk of CRC with regular use of SSRIs. In addition, Chubak et al. [71] reported a not statistically significant trend of reduced CRC risk following antidepressant use. There were no differences in relative risk comparing SSRIs or TCAs users.

About TCAs, the findings of Walker et al. [72] suggest that TCAs may have potential for prevention of both colorectal cancer and glioma. The same group of authors in a following paper [73] assessed the survival of CRC patients who used TCAs post-diagnosis: the authors were not able at this time to find any significant mortality reduction for CRC patients.

Apparently, rather than a decrease of risk, the opposite was demonstrated when TCAs were prescribed for management of cancer pain [72,73].

Lee et al. [74] established that higher cumulative dose of mirtazapine was associated with lower risk for CRC.

More recently, these results have been countered by some conflicting data that have not mirrored the decrease in risk of CRC development shown in previous studies. Indeed, Cronin-Fenton et al. [75] reported in their study that users of TCAs, SSRIs or other antidepressants did not experience any notable reduction on CRC risk. Furthermore, Haukka et al. [76] and Boursu el al. [77] did not find clear evidence of beneficial or harmful association between usage of antidepressant and different types of cancer, including CRC.

In 2018, Kiridly-Calderban et al. [78] assessed the risk of antidepressant use and risk of colorectal cancer in postmenopausal women. The authors included adjustment for depressive symptomatology and were not able to demonstrate a significantly lowered risk of CRC. However, while TCA users had a slight reduction in CRC incidence, it must also be noted that severe depressive symptoms were independently associated with a 20% increased risk of colorectal cancer, confirming the potential role of depression as a confounding factor in other analyses.

Finally, about the risk of recurrence, a large cohort study of stage I to IIIA colon cancer patients was conducted by Pocobelli et al. [79] and did not show an association between use of antidepressants after colon cancer diagnosis and risk of recurrence.

The studies that focused on antidepressants and their role on colorectal cancer patients are summarized in the following Table 3.

### 2.5. Metformin

In the last decade, epidemiological and clinical evidence have shown how insulin resistance and hyperinsulinemia are implicated in the development of different types of cancer, including CRC [80]. Several antidiabetic medications such as insulin, sulfonylureas, dipeptyl peptidase (DPP) 4 inhibitors and glucose-dependent insulinotropic peptide (GLP-1) analogues have been shown to increase the risk of different cancers in diabetic patients [81]. Conversely, multiple epidemiological studies have demonstrated an association between metformin and reduced cancer incidence and mortality. Metformin is an oral biguanide commonly used in treatment of type 2 diabetes mellitus as a glucose-lowering agent which decreases hepatic gluconeogenesis and improves insulin sensitivity by increasing peripheral glucose uptake and use.

In the past years, several studies have investigated the molecular mechanisms of metformin in cancer. On the whole, metformin can induce anticancer activity through two main routes: a direct mechanism resulting from the inhibition of mitochondrial complex I and consequent suppression of adenosine triphosphate (ATP) production in prenoplastic and neoplastic cells and an indirect mechanism related to metformin insulin-lowering activity, which may slow tumour development in hyperinsulinemic patients [82]. In particular, metformin has been shown to modulate AMP-activated protein kinase (AMPK) activation via LKB1 with consequent inhibition of the mTOR pathway, which is crucial in cell growth and proliferation in CRC as in other cancer types.

Despite these intriguing preclinical bases, the sum of available clinical data concerning metformin role as a chemopreventive agent are relatively poor:

Zell JA et al. [83] recently reported the results of a phase IIa trial of metformin for CRC risk reduction in patients with elevated BMI and positive history of colorectal adenomas. Thirty-two patients were treated, with a baseline BMI of 34.9. The authors were not able to find any difference between patients treated with at least 12 weeks of oral metformin and rectal mucosa levels of Ki-67 or pS6 (these two markers were previously proven to be early markers of cancer development).

In a previous study by Higurashi T et al. [84], the authors reported the results of a randomized phase III trial of chemopreventive use of metformin to reduce risk of colorectal adenoma or polyps in post-polypectomy patients without diabetes; 151 patients were enrolled and randomized in a 1:1 fashion to receive metformin (250 mg/daily) or placebo for 1 year. The authors observed a significant reduction of the whole number of polyps found between the two cohorts of patients (RR: 0.67, 95% CI: 0.47–0.97), and a reduction in adenomas was seen (RR: 0.60, 95% CI: 0.47–0.97). No malignant lesions were identified during the course of the study.

Data concerning potential uses of metformin as an adjuvant treatment in patients who have already developed colorectal cancer are even more scarce: while a series of published papers suggest that metformin use might be associated with improved survival [85,86,87,88], in all these studies, the number of patients with unresected metastatic disease is rather low (usually lower than 3–4% of the whole patient population).

To conclude, metformin use might be associated with improved survival for patients with colorectal cancer, but its impact in patients with mCRC has still not been proven.

### 2.6. Statins

Hydroxymethylglutaryl coenzyme A (HMG-CoA) reductase inhibitors, called statins, are common drugs used for decreasing serum total and low-density lipoprotein (LDL) cholesterol, which act by blocking the conversion of HMG-CoA to mevalonic acid.

Statins inhibit also the biosynthesis of several intermediates of the mevalonate pathway, called isoprenoids such as farnesylpyrophosphate (FPP) and geranylgeranylpyrophosphate (GGPP) [89,90]. These isoprenoids have a key role in the posttranslational modification of small GTP-binding proteins, such as Ras, Rac and Rho, involved in the regulation of tumour cell proliferation, cancer progression and activity of immune cells [91,92,93,94]. Recently, it has been demonstrated in in vivo models that statins, by inhibiting the geranylgeranylation of small GTPases, such as Rab5, enhance antigen presentation in antigen-presenting cells and T-cell activation [95], thus showing potential adjuvant properties with cancer immunotherapies.

Other preclinical works have also recognized statins as antitumour and proapoptotic agents in CRC cell lines in vitro and in tumour xenografts [96], yet evidence for the role of statins in the primary prevention of CRC is still conflicting.

In 2005, The Molecular Epidemiology of Colorectal Cancer study, a population-based case–control analysis of the incidence of CRC in northern Israel [97] conducted by Poynter et al., showed a strong inverse association between the risk of colorectal cancer and statin use, leading to a 47% reduction in the risk of CRC, after adjustment for other known risk factors. Conversely, several observational studies did not find an association between use of statins and CRC risk [98,99,100].

A meta-analysis conducted by Bonovas S. et al. included 18 works, either randomized controlled trials (RCTs) of statins for cardiovascular outcomes or observational studies (case control or cohort) and analysed statin use and CRC risk [101]. Unlike Poynter et al., the results of the analysis excluded a potential protective effect of statin use: when the study was restricted to retrospective case-control studies, statin use was significantly associated with CRC risk reduction (around 8%), showing a potential weak protective role. However, when the analysis was conducted only including prospective cohort studies and RCTs, there was no association between statin use and risk of CRC. This modest reduction of CRC risk associated with statin use was further confirmed by the two meta-analyses by Liu Y et al. [102] and Lytras et al. [103].

In 2011, Lee JE and colleagues from Brigham and Womes’s Hospital and Harvard Medical School investigated the association between statin use and CRC risk according to molecular subtypes [104]. In this prospective study, they showed no significant association between statins and overall risk of CRC; however, a potential protective role was found for rectal cancer.

More recently, other cohort studies highlighted the correlation between statin use and reduced CRC risk. The first analysed the non-elderly adult US population (age: 18–64 years) and demonstrated a significant association between statin use and reduction of CRC risk mostly in the younger subgroup (<55 years) [105]. The other one investigated this association in a specific patient subgroup affected by inflammatory bowel disease (IBD). At multivariate analysis, statin use was independently and inversely associated with CRC incidence (OR: 0.42, 95% CI: 0.28–0.62) [106]. Conversely, a recent case-control study performed in Catalonia showed no significant decrease of CRC risk associated to any statin exposure. When stratification was performed by cancer site, reduction of rectal cancer risk was seen in statin users (OR: 0.87, 95% CI: 0.81–0.92) [107]. Overall, the role of statins in relation to the incidence of CRC seems to be still controversial and, to date as data suggest, only a weak action as a chemopreventive agent [108].

Similarly, no consensus has been reached regarding the prognostic effect of statins on CRC. The most recent and updated meta-analysis involving 130,994 patients reported an association between pre-diagnosis statin use and reduced risk of CRC mortality (pooled HR: 0.85, 95% CI: 0.79–0.92). Similarly, an advantage in terms of all-cause mortality (ACM) was observed for post-diagnosis statin use (pooled HR: 0.86, 95% CI: 0.76–0.98). However, when patients were stratified by K-RAS mutational status, no difference in ACM was reported [109].

Studies that have focused on incidental statin use in patients with colorectal cancer can be found in the following Table 4.

Albeit these results are promising, it is worth noting that the effects reported in some of these studies might reflect incomplete control for stage at diagnosis and other confounding factors associated with statin use [110]. The observed associations were weak in magnitude and, especially in post-diagnosis studies, statistical heterogeneity was high [109]. In addition, the few randomized prospective trials in which statins were added to standard chemotherapy were negative [111,112].

Nevertheless, a role of statins in CRC treatment cannot fully be ruled out. There could be some subgroups of CRC patients that could benefit from statin use. The introduction of the “consensus molecular subtypes” (CMS) classification in CRC provided a new fundamental prognostic and predictive tool [113]. CMS2 and CMS3 are considered “cold” tumours, with low immune cell infiltration and downregulation of MHC class I molecules, which results in reduced presentation of tumour-associated antigens [114]. Since statins proved to have vaccine adjuvant proprieties [95], their use in association with immunotherapy might then represent an intriguing option in these specific CRC subtypes. Future trial designs should mandatorily take into account statin type (hydrophobic/lipophilic), dose and therapy duration along with the tumour molecular subtype to better understand statins’ roles in affecting both incidence and outcome of CRC.

## 3. Conclusions

In the last decades, survival for colorectal cancer patients has seen a steady improvement in spite of its high incidence worldwide [115]. This can be explained by advancements in adjuvant therapy and by earlier diagnoses: adjuvant treatment options have recently been finely “tuned” based on patients’ clinical characteristics such as stage of disease involvement [116]. Indeed, earlier stage at diagnosis, favoured by the introduction of screening programmes in several countries, seems to still be the most important factor that determines differences in survival outcomes.

On the other hand, a considerable number of patients who usually do not fit into screening programmes are actually been diagnosed with colorectal cancer. These fit into mainly two categories: younger patients aged less than 40–45 years old and older patients aged 70–75 years old or more. Particularly in older patients, due to frequent comorbidities, management is usually far more complex than in younger people. On the other hand, usually younger patients have higher disease stage upon diagnosis compared to older patients.

Compared to other high prevalence tumour types such as breast or lung cancer, palliative treatment for patients with metastatic colorectal cancer has been usually based on various combinations of chemotherapy. If we exclude the select few patients that have microsatellite unstable tumours and that might benefit from immunotherapy, targeted agents commonly used in colorectal cancer treatment fall within 2 main categories: the anti-EGFR or anti-VEGF pathway. Indeed, the newest addition is the one represented by B-raf inhibitors for patients harbouring B-raf V600E mutations [117] and we could still argue that it is still based on inhibition of a crucial target of RAS-RAF-MEK-ERK pathway. In other cancer types, a relevant number of different treatment options such as immunotherapy with checkpoint inhibitors; PARP inhibitors for patients with impaired homologous recombination system, PIK3CA inhibitors; and tyrosine kinase inhibitors for ALK-, ROS1-, EGFR- and MET-mutated patients have been introduced in the last 5 years for these diseases.

Focusing on differential outcomes for patients who are on incidental treatment for other chronic conditions may allow us to identify a promising group of drugs that would have to be tested in the setting of a clinical trial to discover their effect on cancer specific survival. Toxicities related to treatment would be already well-known due to the pre-existing well-documented use of these drugs in other clinical situations. In addition to that, for some select few groups of patients who are usually underrepresented in clinical trials as very young or old patients, these studies might offer perhaps the only chance to assess the benefit coming from drugs used for treatment of other medical conditions.

Our review has focused on some of the most commonly used drugs: antihypertensive drugs, NSAIDs, antidepressants, metformin, statins and antibacterial antibiotics were covered by our literature research. Taken together, there is a lot of studies that have been published on this matter. However, there is also a lot of heterogeneity between different studies that can lead to wrong assumptions:

From a preclinical point of view, almost all drugs that we have assessed have some form of plausible biological explanation concerning their impact on cancer survival. Furthermore, there is an increasing body of evidence that point to a potential role of incidental drug use as either protective or a risk factor for colorectal cancer: a summary of these studies can be found in Appendix A.

Beta blockers, statins, metformin and ASA seem to have relevant anticancer properties in preclinical models through a series of innovative pathways, related mainly to different proto-oncogene pathways: interference in the c-MYC pathway or different regulations of immune system has been described in these studies. Based on these results, they would be ideal “partners” to be coupled with immune-based treatment options. The other antibiotics seem to have a predominantly negative impact, mainly disrupting the network of signals between the gut microbiota and patient’s immune system.

Despite these promising preclinical data, clinical validation of these drugs still lacks adequate evidence or has only low-quality evidence supporting it. Most of the studies are retrospective case-control or cohort studies, and many others are larger registry studies where important stratification factors that might influence patients’ survival such as tumour sidedness and K-ras/B-raf/N-ras mutational status or, in some cases stage, at diagnosis are taken into account only partially. Furthermore, there is very low evidence in patients with metastatic involvement and who are candidate to receive systemic palliative treatment: in this selected group of patients, where the need for novel treatment options is most dire, data about the use and potential advantage/disadvantage of these drugs is particularly lacking.

If we consider that usually most chronic conditions, such as hypertension, have a relatively late age of onset, it is crucial to analyse the results of the studies presented by taking into account patients’ age as a confounding factor: interestingly, in most studies, differences in antihypertensive drug use due to age itself were not able to reduce a statistically significant impact on overall survival when multivariate analysis was conducted:

In the study of Cui [14], 3-year survival rates of patients aged 50–59 years old was about 75% compared with 70% of patients aged less than 50 and with 63% of patients older than 60 years. As expected, the percentage of patients who actually received antihypertensive medications was higher in patients aged more than 60 years old (65%) compared to younger patients. Despite these differences, survival outcomes for patients who received beta blockers were still better compared to other drugs or to patients who did not receive any antihypertensive medication, independently from age of colorectal cancer diagnosis.

In the study of Jansen [15], among the 4 prespecified age groups, there were statistically significant differences in terms of incidental beta-blocker use, with more than 50% beta-blocker users in the elderly. However, when age-adjusted overall survival analysis was conducted by taking into consideration incidental beta-blocker use, the overall survival benefit of beta-blocker use was maintained. The effect was stage-specific for stage IV patients, thus suggesting that, particularly in more advanced stages of disease involvement, the addition of the drug might determine an additional palliative effect.

Looking specifically at the studies that have been conducted in stage IV colorectal cancer, both Giampieri [16] and Fiala [17] had patient populations where the percentage of elderly patients aged more than 70 years old was around 20% in both studies. In these two studies, there were no differences in age among incidental beta-blocker users or not: on this basis, it cannot be estimated whether the supposed antineoplastic effect of beta blockers might be influenced by patients’ age. It must be noticed that both studies only enrolled in the analyses patients who had received first-line palliative chemotherapy for metastatic colorectal cancer with chemotherapy doublet +/− bevacizumab. Age might have had less impact in these analyses as they only have included elderly patients that were however deemed sufficiently fit to be able to tolerate this kind of chemotherapy and that should be much more similar in terms of fitness to younger patients.

Statins are another class of drugs where adjustment for age usually is required due to the fact that ischemic heart disease is usually but not necessarily more frequent in older patients. Looking at the studies that have been included in this review, only a few have focused on stage IV patients that were actively treated with chemotherapy: in the study of Hoffmeister [110], even though about half of the enrolled population received chemotherapy, only 13.8% of the study population had stage IV colorectal cancer. In this selected group of patients, only 11% actually received statins. When adjustment was performed for sex, age and stage, no differences in overall survival and colorectal cancer specific survival were seen while there was a statistically significant association for statin use and decreased probability of higher stage at diagnosis. This suggests that statins should be tested, rather than in more advanced stages of disease involvement, in earlier stages as they might have some protective effect. The study of Krens [111] was focused on a population comprised entirely of stage IV patients: the authors were not able to prove that statin use was able to determine differences in survival in patients enrolled in the CAIRO-2 trial and who had received palliative first-line chemotherapy with CAPOX + bevacizumab + cetuximab. The biological rationale was based on the fact that statin use would determine changes in prenylation of KRAS, the main factor determining resistances to cetuximab. The study was however negative, and the authors commented on the fact that the severe prognosis of stage IV KRAS-mutated patients might have impaired any effect that could be achieved by statins.

As it can be seen, whenever metastatic disease was compared to the early stage setting, there was an evident change in the impact of incidental drug use. Incidental drug use is markedly different based also on different patients’ ages, thus leading to difficult interpretation of results presented by all these studies. Even with these limitations, from our review, we were able to identify a series of drugs such as beta blockers that should be taken into account as “companions” to new treatment options for metastatic colorectal cancer patients. Other drugs such as statins, antibacterial antibiotics directed against *Fusobacterium* spp., might instead be much more useful in less advanced stages due to their greater effect in reducing overall risk of colorectal cancer development.

We believe that future analyses that will focus on these specific topics will finally manage to answer the question of whether drug repurposing might represent a reasonably useful tool to identify novel candidate drugs for colorectal cancer prevention or treatment.

## Figures and Tables

**Table 1 cancers-12-02724-t001:** Summary of studies for antihypertensive drugs as a prognostic factor.

Drugs	Number of Patients	Results Favouring Drug	Results Not Favouring Drug	Results Against Drug	Citation No.
All antihypertensive	2891	Better OS and DSS for ARB users	No difference in OS and DSS for other antihypertensive drugs	/	Cui [14]
		Better OS and DSS for beta-blocker users			
Beta blockers vs. others	1975	Better OS and DSS for stage IV patients	No difference in OS between beta-blocker users vs. not	/	Jansen [15]
	256 stage IV				
Beta blockers vs. others	8100	/	No difference in OS between beta-blocker users vs. not	Long-term use related to higher mortality	Jansen [16]
Beta blockers vs. others	235 stage IV	Better OS and PFS for patients treated with chemotherapy	No difference in OS and PFS for patients treated with bevacizumab	/	Giampieri [17]
Beta blockers vs. others	514 stage IV Bevacizumab treated	Better OS and PFS for beta-blocker users	/	/	Fiala [18]
ARB/ACEi users vs. others	1511	/	No difference in OS vs. patients treated with other drugs	/	Cardwell [19]

**Table 2 cancers-12-02724-t002:** Summary of studies for FANS/non-steroid anti-inflammatory drugs (NSAIDs) as a prognostic factor.

Drugs	Number of Patients	Results Favouring Drug	Results Not Favouring Drug	Results Against Drug	Citation No.
ASA	In vitro	Inhibition of cancer cell proliferation	\	\	Mitrugno [38]
Juglone, Indometacina	In vitro	Reduction of the occurrence of CRC	\	\	Seetha [40]
ASA	In vitro	Enhancement of the cisplatin-mediated inhibitions of cell proliferation, migration and invasion and the induction of apoptosis in CRC	\	\	Jiang [42]
ASA vs. not users	66	Improvement of the clinical outcome of heavily pretreated patients with metastatic colorectal cancer receiving chemotherapy	\	\	Giampieri [43]
ASA vs. not users	241	Anticancer activity against rectal cancer during preoperative CRT	\	\	Restivo [44]

**Table 3 cancers-12-02724-t003:** Summary of studies for antidepressants as a prognostic factor.

Drugs	Number of Patients	Results Favouring Drug	Results Not Favouring Drug	Results Against Drug	Citation No.
TCA vs. others	16,519 CRC	/	Improved survival for TCA users in glioma	Decreased survival for TCA users in CRC	Walker [73]
	(+1364 gliomas)				
SSRI vs. nonuser TCA vs. nonuser	1923 resected CRC stage I-IIIA	/	No difference in CRC risk of relapse between SSRI or TCA users vs. nonuser	/	Pocobelli [79]

**Table 4 cancers-12-02724-t004:** Summary of studies for statins as a prognostic factor.

Drugs	Number of Patients	Results Favouring Drug	Results Not Favouring Drug	Results Against Drug	Citation No.
Statins vs. nonusers	Meta-analysis of 14 studies (130,994 patients)	Both pre-diagnosis and post-diagnosis, statin users have reduced all-cause mortality and cancer-specific mortality	/	/	Li [109]
Statins vs. nonusers	2697 patients of whom 412 statin users (cohort study)	/	No association between statin use and overall, CRC-specific or recurrence-free survival	/	Hoffmeister [110]
Statins vs. nonusers	529 patients of whom 78 were statin users (post hoc analysis from a phase III randomized controlled trial)	/	No association between statin use and overall and progression-free survival in KRAS mutant metastatic CRC patients treated with capecitabine, oxaliplatin bevacizumab ± cetuximab	/	Krens [111]
FOLFIRI/XELIRI plus simvastatin (40 mg) vs. FOLFIRI/XELIRI plus placebo	269 patients (phase III randomized controlled trial)	/	No improvement in terms of progression-free and overall survival by the addition of simvastatin	/	Lim [112]

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
