# Peer review of "Impact of Polypharmacy for Chronic Ailments in Colon Cancer Patients: A Review Focused on Drug Repurposing"

_cancers, 2020, doi:10.3390/cancers12102724_

Round 1
Reviewer 1 Report
This is an interesting study which tries to summarize the many studies which have correlated colorectal cancer occurrence in patients involved in prospective trials and not.
As usual, there is the possibility of errors, because most of the studies were perfomed for pirmary prevention or treatment of diseases different from colorectal cancer, and one may wonder how accurate might be the diagnosis of colorectal cancer if this was not a primary outcome in the study.
Meta ananalyses have directed primarly to assess the efficacy of aspirin (at different dosages) given as primary prevention of cardiovascular events . The studies have generated controversies : aspirin seems to have effect only after 10 years in studies performed before 2000, whereas in studies performed after 2000 there was no effect of aspirin in colorectal cancer prevention.
There is the possibility that this difference might depend on the introduction into clinical practice of many drugs (new more effective antibiotics, new statins better tolerated by patients, new anti-hypertensive drugs etc etc) which in general are not reported in detail even in prospective randomized studies performed in high level academic centers.
We are sure only about the fact that since 1980 (almost 10 years after the first reports about the efficacy of aspirin in preventing cardiovascular events) te incidence of colorectal cancer in several western countries is declinig at a rate of 2.1% per year with reduced mortality rates.
We know that chronic inflammation may favour cancer occurrence and progression either at a local or central level (immuno-derugulation) , and we might hypothesize that this decerased incidence could be related to reduced chronic inflammation in the general population. However, it is difficult to determine which specific drug has a major role in reducing chronic inflamamtion in the general population, in a scenario where drugs with known or unknown (for example statis at the beginnign) anti-inflammatory effect are continuously introduced and substituted by new drugs.
Having accepted the logical and obvious limitations of any study trying to define the effect of one drug at the time, the study has the obvious interest to
give some kind of definition about the possibility that specific drugs might prevent colorectal cancer.
The majority of the drugs analyzed in the study (anti-hypertensive drugs, statins, aspirin) are prescribed to older patients in whom we are documenting a decrease rate of colorectal cancer, but not to younger patienst in whom we are documenting an increased rate of colorectal cancer in the same countries.
The question is which drug is involved , if any?
To give a correct answer is impossible with analyzing the data so far available.
Probably these matters should be included in the discussion.
I have few other suggestions:
1-A table including the studies analyzing the effect of aspirin on colorectal cancer prevention should be added.
2-A separate table should be included for studies in which the primary outcome was the occurrence of colorectal cancer
3-If possible a table should be included of studies in which the levels of C Reactive Protein have been reported in patients developing colorectal cancer and not and how the examined drugs acted on CRP levels.
Minor observations:
-The study of Litras seems to describe a positive effect of statins in preventing colorectal cancer either in meta analyses or retrospective studies (I am not sure)
-The study of Hoffmeister seems to describe a cooperative effect between aspirin and statins.
I think the study is important. I would reccomend to analyze only few drugs at the time in more detail. For instance the analysis of anti-hypertensive drugs on colorectal cancer occurrence is very interesting, and it should represent a single study analyzing all the reports in more detail, specifically which drugs have been added to the antihypertensive agent and how the levels of CRP have changed. Similalry the role of aspirin and other non steroidal drugs should be analyzed in a seprate study, eventually adding analysis of the effect of statins, with or without the simulteneous administrtaion of aspirin.
I think that the objective of the study is very timely and important.
I would to reccomend to divide the paper in two papers, one describing the effect of antihypertensive drugs and one describing the effect of non steroidal antiinflammatory drugs and statins.
Reviewer 2 Report
- The present kind of review can sometimes be a little hard to read as being an enumeration of trial and drugs. I welcomed the tables at the end as summarising the different studies in a clear manner. Therefore I think that the manuscript could gain clarity if the authors would introduce a shortened table in the main text. Some expressions such as "No difference in CRC risk between TCA users vs other" could be easily reduced to fit in two lines.
- Some transition sentences are missing between the different paragraphs.
- Please avoid starting sentences with repeating expressions such as our study suggests.
- The authors also use a lot of abbreviations that reduce the readability of their paper especially as the abbreviations outside the parenthesis that are not always written out in the text such OS, CRC, etc. Inside the parenthesis, I understand that not all the abbreviations in the statistics need an explanation but if used outside they must be explained at least once.
- Further, as the text contains a lot of parentheses due to the statistics, I would remove all parentheses that are linked to your writing style like in this example: These drugs have been extensively used in patients with colon cancer, particularly when patients are receiving anti-VEGF based chemotherapy combinations (another factor leading to increased blood pressure).
- From a format point of view, the statistics should be consistent. In some cases, the CI (for confidence Interval) or OR( for odds ratio) is present in the parenthesis sometimes not, and sometimes there is followed directly by the value and there is some punctuation.
- Replace head&neck by head and neck.
- The introduction only cited 2 papers. Some references could be added i.e. for the examples of amantadine and thalidomide.
- Is “ad diagnosis” a typo or an abbreviation
- please be consistent with the references. Sometimes the punctuation is before the ref and sometimes after.
- please check the paper for typos, missed punctuation, lack consistency
- I would also a longer discussion that critically accessed the different studies. Why are there some discrepancies?
Round 2
Reviewer 1 Report
This is a very timely and interesting review.
As stated by the Authros and by the reviewers, the many hidden and not-hidden variables do not allow defintive conclusions or hypotheses.
The review can be useful to the many researchers involved in this field.